# Epidemiological Investigation of Pediatric Fractures—A Retrospective Cohort Study of 1129 Patients

**DOI:** 10.3390/medicina59040788

**Published:** 2023-04-18

**Authors:** Xiaoliang Xiao, Yuhong Ding, Yiqiu Zheng, Yun Gao, Huaqing Li, Ruikang Liu, Ruijing Xu, Pan Hong

**Affiliations:** 1Department of Orthopaedic Surgery, Zhuhai Center for Maternal and Child Health Care, Zhuhai 519000, China; 2Second Clinical School, Tongji Medical College, Huazhong University of Science and Technology, Wuhan 430074, China; 3Basic Medical School, Tongji Medical College, Huazhong University of Science and Technology, Wuhan 430074, China; 4Department of Endocrinology, Union Hospital, Tongji Medical College, Huazhong University of Science and Technology, Wuhan 430074, China; 5Department of Orthopaedic Surgery, Union Hospital, Tongji Medical College, Huazhong University of Science and Technology, Wuhan 430074, China

**Keywords:** children, fractures, incidence, public health, policy

## Abstract

*Background and Objectives*: Fractures are common in pediatric trauma, and they are caused by a broad spectrum of factors. Only a few studies have discussed the mechanisms of injury and their relationships to different types of fractures. The most frequent type of fractures in different age groups remains unclear. Therefore, we aim to summarize the epidemiological characteristics of pediatric fractures in a medical center in Zhuhai, China from 2006 to 2021 and analyze the causes of fractures with the highest frequency in different age groups. *Materials and Methods*: We extracted the information from the Zhuhai Center for Maternal and Child Health Care of those under 14 years old who had fractures from 2006 to 2021. *Results*: We reviewed the information of 1145 children. The number of patients increased during the 15 years (*p* < 0.0001). The number of patients was significantly different between genders after Y2 (*p* = 0.014). In addition, more than two-thirds of patients (71.3%) had upper limb fractures, and all types of falls were the most common cause of fractures (83.6%). The incidence demonstrated an insignificant difference in age groups except for the fractures of humerus and radius. Moreover, we discovered that the prevalence of fall-related injuries decreased with age, while that of sports-related injuries increased with age. *Conclusions*: Our study demonstrates that the prevalence of fall-related injuries decreases with age, and that of sports-related injuries increases with age. Most patients have upper limb fractures, and all types of falls are the most common cause of fractures. Fracture types with the highest frequency differ in each age group. These findings might supplement current epidemiological knowledge of childhood fracture and provide references for decision-making in children’s health policies.

## 1. Introduction

Childhood injuries account for more than 10 million annual hospital visits in the United States, making them the second leading cause of visiting primary healthcare facilities and the emergency departments of hospitals [1]. Fractures comprise 10–25% of musculoskeletal injuries and all pediatric injuries [2,3]. For injured children, on average, arm fractures restricted the patient’s activity for 14 days (95% CI 8–20 days) and leg fractures for 26 days (95% CI 7–45 days) [4]. In addition, it is noteworthy that the high incidence of fractures in one’s childhood is related to the individual higher risk of fracture in adulthood [4]. Korula et al. reported that 50% of children would sustain a fracture before 18 years old, and 20% of children would suffer two or more instances of fracture [5]. Therefore, the epidemiological study of fractures in children is imperative.

Previous studies have summarized the epidemiological characteristics of fractures in children. The upper extremity is the most frequently injured region [3,6,7,8,9,10]. In addition, fractures result from various factors across different age groups. A study published by Hedström et al. highlighted the danger of children’s activities as an essential factor in determining the incidence of fractures [11]. Wang et al. demonstrated that injuries resulting from motor vehicle collisions decreased with age in children, while injuries from being hit by others and sprains increased with age [12]. To the best of our knowledge, there was no study discussing the mechanisms of injury and their relationship to different types of fractures. Overall, the most common type of fractures in different age groups remains to be elucidated.

Zhuhai is the first batch of special economic zones in the Guangdong Province of the People’s Republic of China, which is characterized by its particular economic development strategy and high population growth rate in recent years. Zhuhai’s per capita Gross Domestic Product (GDP) reached 145,600 Yuan (20,997.98 USD) in 2020, ranking first in China, and the ten-year growth rate of the permanent population ranked second in Guangdong province [13,14]. In addition, the permanent resident population of Zhuhai increased from 1,566,200 (2010) to 2,439,600 (2020), and 15.88% of the population were children aged 0–14 years [13].

We collected and reviewed all the patient information registered in the department of pediatric orthopedics, Zhuhai Center for Maternal and Child Health Care, from 2006 to 2021. We summarized the epidemiological characteristics of pediatric fractures over 15 years. We demonstrated the fractures with the highest frequency in each age group and analyzed the direct causes of fractures. Moreover, we performed subgroup analyses of incidences by age and sex, respectively. These findings might supplement current epidemiological knowledge of childhood fracture and provide references for decision-making of children’s health policies.

## 2. Materials and Methods

We performed a retrospective cross-sectional study consisting of 1145 inpatient children under 14 years old in Zhuhai Center for Maternal and Child Health Care, China from 2006 to 2021. We defined exposure as the direct injuries and outcome as the occurrence of fracture. Original data were collected from the electrical medical record system of the Zhuhai Center for Maternal and Child Health Care. Eligibility criteria included being admitted to Zhuhai Center for Maternal and Child Health Care, being a child (under 14 years old), and being diagnosed with a fracture in any part of the body. All patients underwent rigorous physical examination and medical history questioning to rule out the possibility of child abuse. In our study, no patient with signs of non-accidental injuries (NAI) was found. Therefore, NAI data referring to hitting, kicking, burning, biting, or choking were not included. Medical record numbers were used to ensure that every child was recorded only once for each injury. The direct causes of injuries were recorded according to statements from the children and parents. Moreover, we recorded each fracture separately, while multiple vertebrae, toe, and finger fractures were recorded as one. Two researchers, PH and YG, reviewed the charts and radiographs of all patients independently to verify the fracture diagnosis.

### 2.1. Data Extraction

The medical data of 1145 patients admitted to the Children’s Orthopedic Ward of the Zhuhai Center for Maternal and Child Health Care record during the 15 years were included. The study extracted the following parameters: age, gender, admission date and discharge date, fracture diagnosis, and cause of injury. Repeated data, data containing missing values, and data unrelated to fracture were excluded. Two researchers, YD and RL, extracted the data separately, and discrepancies were checked and resolved by the third author, PH. The patient flowchart is shown in Figure 1.

### 2.2. Statistical Analysis

Data were exported to R programming language (version 4.2.1) for analysis which YD conducted, and RL reviewed the analysis process. Descriptive statistics of percentage (one significant digit after the decimal point was kept) and median were employed. Welch Two Sample *t*-test and Mann–Withney–Wilcoxon test were used to compare the incidence of specific fractures between different age, gender, and cause of injury groups. The Mann–Kendall trend test was applied to obtain the change of trend of fracture incidence. *p*-values < 0.05 were considered statistically significant (two-sided).

To analyze more conveniently, we abbreviated the original types of fractures and causes of the injuries as codes. There are 11 causes included in the study: Birth Injury (BI); excise machine injury (EM), referring to falling with greater momentum involving sports equipment referring to skating, skiing, rollerblading, skateboarding or bicycle riding, etc.; falling from one plane to another (FPA), referring to bed, chair, sofa, stairs or steps, etc.; falling on the same plane (OSAP), referring to slip, trip, etc., with small momentum; improper use of force (IUF), referring to repetitive motion or overexertion, etc.; mechanical force (MEF), referring to exposure to tools and instruments involving knives, glass, door gap, falling object, or lifting device, etc.; cyclist injury (CY), referring to bicycle collision accident except collapsing with a car; motorcycle passenger injury (MP), referring to passengers or pedestrians in motorcycle accidents; vehicle passenger injury (VP), referring to passengers or pedestrians in vehicle accidents; physical contact with people (PCP); other unexpected reasons (OUR). In addition, there are 17 types of fractures included in the study: clavicle fracture (CLF), growing skull fracture (GSF), humerus fracture (HF), ulna fracture (UF), radius fracture (RAF), fracture of radius and ulna (RULF), phalanx fracture (PF), pelvic fracture (PEF), femur fracture (FEF), patella fracture (PAF), fibula fracture (FIRF), metacarpal fracture (MF), tibial fracture (TIBF), tibia and fibula fracture (TAFF), foot fracture (FF), and whole-body fracture (WB).

### 2.3. Ethics Approval

The study was conducted under the Declaration of Helsinki, and informed consent was waived and approved by the Ethical Committee of Wuhan union hospital. No private information of patients who participated in this study was needed.

## 3. Results

### 3.1. General Information

We recorded information on all 1145 children (≤Y14) who were treated for a fracture at Zhuhai Center for Maternal and Child Health Care, China from 2006 to 2021. Two pieces of repeated data and fourteen pieces of data unrelated to fracture were excluded. The final data were extracted from 1129 children (458 girls and 671 boys) meeting the criteria. Ages ranged from newborn to 14 years old, and 984 patients (87.2%) were local citizens of Guangdong. All data provided basic information about age, gender, admission date and discharge date, fracture diagnosis, and cause of injury. Other data such as medical record numbers, operation methods, surgery costs, home addresses, etc., have been excluded because they were unrelated to our research purpose. Radiographs confirmed all fractures. No follow-up information was recorded in the inpatient record system.

### 3.2. Epidemiology

Figure 1 illustrates the annual number of pediatric patients between 2006 and 2021. The annual number of patients was on the rise in general and surged in 2015 (107 cases). However, it plummeted in 2010 (17 cases), 2016 (79 cases), and 2019 (91 cases). The number reached a peak in 2021 (185 cases).

Figure 2 shows the gender ratio and length of hospital stay. In total, 1129 patients comprised 671 males (59.4%) and 458 females (40.6%). The numbers of patients had no significant difference between genders until Y3 (*p* = 0.014). In terms of the mean duration of hospital stay, no significant difference was found between males (10.8 days) and females (10.6 days).

Figure 3 shows the age distribution of children treated for fractures during the 15 years. The number of patients was concentrated in Y1 to Y7 (777, 68.8%). The peak was Y2 (150 cases), followed by Y3 (126 cases), Y1 (117 cases), and Y4 (109 cases).

Figure 4 presents the distribution of fracture types in children from 2006 to 2021. The number of patients with HF was the highest (489, 43.3%), while the number of patients with RULF ranked second (121, 10.7%) and was closely followed by other upper limb fractures patients: UF (110, 9.7%) and RAF (85, 7.5%). The numbers of patients with lower limb fractures were FEF (93, 8.2%), TAFF (52, 4.6%), and TIBF (39, 3.5%). All other types of fractures were included in DMER (34, 3.0%), and the corresponding numbers of patients were FF (two cases), WB (three cases), FIRF (five cases), GSF (five cases), MF (nine cases), PAF (two cases) and PEF (eight cases).

Figure 5 presents the distribution of causes in children. The fractures of children were predominantly caused by falls: OSAP (815, 72.2%) and FPA (130, 11.5%), while MEF was less frequent (49, 4.34%). Other remaining causes were included in CMER (22, 1.94%), which consisted of BI (three cases), CY (seven cases), PCP (five cases), and OUR (seven cases).

### 3.3. The Proportion of Different Causes in All Ages of Children

Figure 6 shows the percentages of causes in all ages, and Table 1 presents the specific numbers of each proportion. OSAP was the most common cause of injury, and the proportion of OSAP increased from <M1 (25.0%) to Y1 (70.6%) (*p* = 0.009) but was basically constant in ages ≥Y1 (varied from 60% to 86.3%). In addition, the cause composition of <M1 was unique: BI (37.5%)contributed the most to injury, while OSAP (25.0%) placed second, followed by FPA (12.5%), IUF (12.5%) and OUR (12.5%). As age increased from <M1 to Y14, the proportion of FPA decreased (*p* = 0.003) and stabilized after Y2. However, the proportion of EM increased (*p* = 0.01) and was concentrated in Y8-Y13 (5.8%). Moreover, it seemed that MEF was concentrated in Y2–Y3 (8.3%), and PCP happened with higher frequency in Y13–Y14 (12.1%), but they all had no statistical difference.

### 3.4. The Proportion of Causes in Types of Fractured Children

Figure 7 shows the percentage of causes in fractured children, and Table 2 presents the specific proportion. Patients with upper limb fractures referring to HF (84.7%), UF (78.2%), RAF (95.9%), and RULF (80.2%) were mainly caused by OSAP, while FPA placed second. OSAP took a large portion in children with PAF (50.0%), CLF (64.6%), FEF (52.7%), TIBF (51.3%), and MF (77.8%). As the second cause, FPA led to many kinds of fractures, but most were upper body part fractures, especially GSF (40.0%). However, VP contributed to most of the fractures in lower limbs, including FIRF (40.0%), PEF (50.0%), TAFF (25.0%), and TIBF (15.4%). Severe fracture of the whole body (WB) was also mainly caused by VP (66.7%) and FPA (33.3%). In addition, MEF was only related to distal limbs: MF (22.2%) and PF (65.5%). Moreover, IUF resulted in FIRF (40.0%) and FF (50.0%), while the other reason causing FF was MP (50.0%). MP led to CLF (8.3%), PEF (12.5%), and TAFF (3.8%). As for patients with lower limb fractures, TAFF and TIBF resulted from the relatively greatest variety of causes, including CY, EM, FPA, IUF, OSAP, PCP, and VP. As a particular cause, PCP mainly resulted in GSF (20.0%) and TIBF (2.6%).

### 3.5. Relationship between Gender & Age and Incidence

As for gender groups, the number of patients was significantly different between males and females after Y2 (*p* < 0.01). The incidence of each fracture was consistent between different genders, and there was no significant difference in the annual incidence of each fracture. As for the age group, the incidence of HF decreased gradually after Y8 (*p* = 0.009). However, the incidence of RAF increased, where significant differences existed in the three stages: <Y3, Y4–Y7 and >Y8 (*p* = 0.0003 between <Y3 and Y4–Y7; *p* = 0.001 between Y4–Y7 and >Y8). There was no significant difference in other fractures among different age groups.

## 4. Discussion

Our single-center study summarized the epidemiological data and risk factors for childhood fractures in Zhuhai, Guangdong, from 2006 to 2021. Specific findings needed to be highlighted. (1) The number of patients increased rapidly during these 15 years (*p* < 0.0001). (2) The age of patients peaked at Y2 (13.3%) and was concentrated in Y1–Y7 (68.8%). (3) The number of patients significantly differed between boys and girls after Y2 (*p* = 0.014). (4) As for types and causes of fractures, more than two-thirds of patients (71.3%) had upper limb fractures, and all types of falls were the most common cause of fractures (83.6%). (5) Fall-related injury frequency decreased with age, and sports-related injury frequency increased with age. (6) The incidence of each fracture was consistent between different genders. (7) There was no significant difference in the annual incidence of each fracture. (8) The incidence showed no difference between age groups except for the fractures of humerus (dividing line of Y8) and radius (dividing line of Y3–Y7).

In all, we believe that the causes of fractures are related to children’s growth. Therefore, we managed to explain the epidemiological data and analyze the reasons for each high-incidence fracture at different ages. We hope that these findings could provide epidemiology evidence for childhood fractures and help formulate policies for children’s health.

Firstly, infants are easy to hurt during childbirth. Therefore, BI (Birth injury) ranked as the top cause of fracture (37.5%) in patients <M1 group. A birth-related injury was defined as any traumatic, ischemic event sustained during delivery [15]. Lalka et al. showed that the top five risk factors for BI (Birth injury) were shoulder dystocia, instrumented forceps birth, breech delivery, gestational diabetes, and multiple births [16]. However, birth injuries only occurred from 2007 to 2009, and the incidence decreased from 5.9% (2007) to 3.4% (2009) in our study. It is possible that better birth delivery techniques could explain the improvement. Figure 4 demonstrates that falls (all kinds) were the leading cause of injury, consistent with an earlier study [17]. In the research conducted by Shimony-Kanat et al., falling from furniture was the leading cause of injury in <Y1 (37.9%), and slipping was the leading cause in Y2–Y3 (38.4%) [17]. Obviously, the muscles of ≤Y2 children were not strong enough to support them and move freely, and they easily fell from furniture without protection. In addition, infants lack self-protective reflexes which may result in more severe injuries [18]. Bittle et al. demonstrated the top four risk factors of falls: an infant left unattended, an infant held by a sleeping adult, a presence of less than two side rails up on the bed, and the elevated position of the bed [19]. Moreover, falls mainly resulted in upper body fractures since most falls occurred with the forearms touching the ground first. In contrast, thicker subcutaneous fat could protect the bone of lower limbs.

Their curiosity to explore the world grows as the children grow up. Exploration and active learning are the typical features of development and maturation [20], and children value the exploration of uncertain options highly [21]. In addition, children delight in feeling all things with their hands which makes their hands prone to injury. Therefore, cuts, crush injuries, smashes, and other mechanical-related injuries related to the fingers and palms (15.6%) are more common at Y2–Y3. However, as their age increases, children gain better control of their bodies. Their athletic ability and desire for outdoor sports enhance gradually. Alves et al. summarized physical activity during children’s growth and concluded that the intensity of physical activity increased with better musculoskeletal strength [22]. Therefore, FPA (falling from one plane to another refers to bed, chair, sofa, stairs or steps, etc.) becomes less common after Y2. The risk of CY (bicycle collision accident except colliding with a car) and EM (falling with greater momentum involving sports equipment refers to skating, skiing, rollerblading, skateboarding or bicycle riding, etc.) is associated with exercise increased from 0 to an average of 1.1% (CY, Y2–Y7) and an average of 5.8% (EM, concentrate in Y8–Y13). In addition, PCP (physical contact with people) occurs only in Y13–Y14 children (12.1%), which may relate to aggressive characters at this age. Meanwhile, traffic accidents involving electric cars, motorcycles, and vehicles (3.6%) cannot be ignored. They may be related to local traffic conditions, road planning, and children’s protection equipment.

As for the epidemiological analysis, our results demonstrate that the number of patients increased over the years. First of all, the steady increase in the permanent resident population in Zhuhai, which increased from 1,566,200 (2010) to 2,439,600 (2020) with an average annual increase of 87,900 and an average annual growth rate of 4.57%, is an outstanding factor [13]. As the number of children increased, the number of patients increased with an average annual growth rate of 11.82% from 2010 to 2020. However, we could not obtain the exact number of pediatric population, and we merely observed a positive correlation between the increase in patients and the increase in pediatric population. In addition, we believe that the Two-Child Policy, the growing awareness of medical treatment, and the affordable health cost contributed to the increasing number of patients. In our study, the fracture age peaks at Y2 (13.3%) and is concentrated in Y1–Y7 (68.8%). However, in a study by Mäyränpää et al., children ≤Y16 with fractures in Helsinki were analyzed [2], and the authors demonstrated that the incidence of fractures peaked at Y10 in girls (28.6‰) and Y14 in boys (38.3‰), with another smaller peak at Y5 in both genders. Cooper et al. reported that the incidence of pediatric fractures in the UK peaked at Y11 in girls (10.1%) and Y14 in boys (9.5%) from Y0 to Y17 [23]. The probable reasons for the inconsistency were different population structures, regional climate, and diverse ideas about seeking medical services. In addition, our data were solely collected from a maternal and childcare service, and older children in Zhuhai might choose other local comprehensive tertiary hospitals for treatment.

As for the subgroup of gender, girls are less likely to injure themselves than boys after Y2. Similar conclusions have been reported in previous studies [2,24,25]. Mäyränpää et al. concluded that the risks of fractures of boys (63%) were almost twice those of girls (37%) [2]. We believe that different outdoor activity participation between genders might explain the difference. Ceppi-Larraín et al. observed a significant difference in favor of boys over girls to physical activity in Temuco, Chile [24]; Colley et al. demonstrated that girls generally have lower levels of fitness compared to boys (Y6–Y19) except for flexibility [25]. Therefore, it appears to be the effect of gender difference that boys display higher participation in sports leading to a higher risk of fractures [26].

Supracondylar and distal radial fractures of the humerus remain the top two fractures in childhood. The reason for the high incidence of humerus fracture is controversial. Humerus fractures usually demand higher trauma energy, and often are not at the same level as conventional fall injuries [27]. The study by Pogoreli’c et al. concluded that a fall on an elbow or an extended arm could result in a fractured humerus [28]. Moreover, a study by Ma et al. suggested that sedentary activities such as watching the computer and television might be related to the increased incidence of the forearm fractures in childhood [29]. As for fractures resulting from car accidents, most of them are fractures of the pelvis, femur, fibula, and tibia. We suppose that an upright position of the children at the time of impact might explain the phenomenon, and the limitation of children’s height may lead to the injury of pelvis and femur. Therefore, injury-related reasons seem more important than biological and anatomy-related reasons [27]. Patel et al. demonstrated that fractures of the radius and ulna are the most common fractures of the upper extremity [30]. In addition, Updegrove et al. summarized the treatments of humerus fractures [31]. It could be concluded that the complex anatomy of humeral fractures leads to various treatments, as these fractures carry the risk of neurovascular compromise [31]. Although conservative treatments of radius and ulna fractures in outpatient service are common, hospitalization of humerus fractures is still generally required [30,31]. Therefore, humerus fractures are frequent in our study.

Other reasons may contribute to the statistical difference: calcium and nutrient intake, obesity, sports and exercise, urban/rural residence, and seasonal differences. Several studies have corroborated some of these factors. Obesity in childhood led to more complex humeral fracture patterns and higher rates of limb fractures than non-obese children [32,33]. Varkal et al. concluded that high body mass index and low Vitamin D intake are related to increased incidence of childhood fractures [34]. Meanwhile, the study by Clark et al. showed that children with low bone mineral density have a much higher risk of fracture throughout childhood [35]. Physical activity acts as an essential factor. Vigorous physical activity leads to increased fracture risk, but reducing physical activity decreases bone strength and results in higher fracture risk [36]. In addition, Lund et al. reported that the incidence of childhood fractures peaks in spring and summer (404 cases in boys and 317 cases in girls in June) [37]. They believe warm weather increases physical activity, resulting in seasonal incidence change [37]. In all, many indirect reasons for fractures mentioned above were not included in our study because they were hardly recorded in the inpatient database.

We have summarized a few tips for primary health care providers. (1) <Y1 children should be placed in bed or sofa, etc., with guardrails which could prevent tipping over and falling. (2) Sharp objects should be placed away from the living area for Y2–Y3 children, and fragile or heavy objects should be kept beyond the reach of children. (3) Children should wear appropriate protective gear during exercise after developing sufficient athletic ability. (4) Child seats should be routinely placed in the car, and helmets are needed while sitting in the back seat of electric cars or motorcycles. (5) As drivers, the guardians should observe the road vigilantly; as pedestrians, the guardians and children should keep paying attention to the road conditions. (6) Parents and teachers should educate the children not to solve problems with mere brute force.

The advantages of our study are the following: (1) the epidemiology data of Chinese children fracture is relatively scarce, and our study has filled a part of the gap. The period of our data (15 years) is long enough to deliver reliable conclusions; (2) it is novel that our study analyzed the most frequent fracture types in all age groups of children; (3) targeted recommendations for children’s health care were provided.

However, there were several limitations in our study. Firstly, we only analyzed the information extracted from the inpatient registry system; indirect cause information was unavailable. Secondly, analyses of more detailed subgroups were not performed. In addition, some potential factors that may lead to bias such as nutrition level and education level were not clarified. In the future, multiple-center studies on the epidemiology of childhood fractures (both inpatients and outpatients) are warranted to help formulate better children’s health policies and injury prevention guidelines for children living in PR China.

## 5. Conclusions

Our study demonstrates that fall-related injury incidence decreases with age, and sports-related injury incidence increases with age. Most patients have upper limb fractures, and all types of falls are the most common cause of fractures. Fracture types with the highest frequency are different in each age group. These findings might supplement current epidemiological knowledge of childhood fracture and provide references for decision-making of children’s health policies.

## Data Availability

The data are not publicly available due to the restrictions of our institution.

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
