# Peer review of "Epidemiological Investigation of Pediatric Fractures—A Retrospective Cohort Study of 1129 Patients"

_medicina, 2023, doi:10.3390/medicina59040788_

Round 1

Reviewer 1 Report

Re: "Epidemiological investigation of pediatrics fractures - a retrospective cohort study of 1142 patients"

Major comment:

1) This study is interesting, but the quality of the data presented should be improved. The Authors should use the "Medicina" template, as the text is bad-organized which influences the quality of the manuscript.

2) Please provide a conclusions section

3) Please provide clearly defined limitations section/paragraph

4) This manuscript should be checked by a native English speaker

Minor comments:

1) Please provide the consecutive numeration of tables, Table 1, 2, 3, etc. rather than 1a, 1b, etc.

2) Please revise the conclusions in the abstract - please provide conclusions that will be in line with the findings obtained by the Authors, rather than "general conclusions", as it is obvious that different fractures may be observed in different age groups.

Author Response

Dear reviewer,

Thank you for your time. Please see the attachment.

Reviewer 2 Report

The authors present an epidemiologic study of the association between mechanisms of injury, age, sex, and type of fracture in children. This study is of interest to the reader because it presents relevant data from an Asian cohort and the authors discuss the differences with European cohorts. The number of patients included, 1142, is large enough to draw reasonable conclusions. I suggest to accept the paper and have only some minor comments and questions:

1. I do not understand "[...] mechanisms of injury and direct causes of fractures together." "[...] mechanisms of injury and their relationship to different types of fractures" might be a better solution.

2. The increase in patients should be normalized to the increase in the (pediatric) population from 2006 to 2021.

3. The many abbreviations sometimes make it difficult for the reader to read the paper quickly.

4. Sometimes words like "Duration," "Cause," or "Other unexpected reasons" are capitalized, for which I see no reason.

L. 324: "C" is missing for clavicle fracture.

Author Response

(The authors gave the same response as above.)

Reviewer 3 Report

please have a native english speaker edit your text before sending out again.

Author Response

(The authors gave the same response as above.)

Round 2

Reviewer 1 Report

The Authors addressed all the comments.

Author Response

(The authors gave the same response as above.)
